# A New Functional Screening Platform Identifies Colistin Sulfate as an Enhancer of Natural Killer Cell Cytotoxicity

**DOI:** 10.3390/cancers14122832

**Published:** 2022-06-08

**Authors:** Serena Cortés-Kaplan, Reem Kurdieh, Mohamed S. Hasim, Shelby Kaczmarek, Zaid Taha, Glib Maznyi, Scott McComb, Seung-Hwan Lee, Jean-Simon Diallo, Michele Ardolino

**Affiliations:** 1Cancer Therapeutics Program, Ottawa Hospital Research Institute, Ottawa, ON K1H 8L6, Canada; scort096@uottawa.ca (S.C.-K.); rkurdieh@ohri.ca (R.K.); mshasim489@gmail.com (M.S.H.); ztaha@ohri.ca (Z.T.); gmaznyi@ohri.ca (G.M.); jsdiallo@ohri.ca (J.-S.D.); 2CI3, University of Ottawa, Ottawa, ON K1N 6N5, Canada; skacz022@uottawa.ca (S.K.); scott.mccomb@nrc-cnrc.gc.ca (S.M.); seunglee@uottawa.ca (S.-H.L.); 3Department of Biochemistry, Microbiology and Immunology, University of Ottawa, Ottawa, ON K1N 6N5, Canada; 4Human Health Therapeutics Research Centre, National Research Council, Ottawa, ON K1A 0R6, Canada

**Keywords:** NK cells, immunotherapy, functional screen, drug repurposing

## Abstract

**Simple Summary:**

The use of small compounds in cancer immunotherapy has been limited so far. Her we screen for drugs that enhanced the ability of immune cells to kill tumor cells and identified the molecule Colistin Sulfate as a booster of immune activity.

**Abstract:**

Due to their crucial role in tumor immunity, NK cells have quickly became a prime target for immunotherapies, with the adoptive transfer of NK cells and the use of NK cell engagers quickly moving to the clinical stage. On the other hand, only a few studies have focused on small molecule drugs capable of unleashing NK cells against cancer. In this context, repurposing small molecules is an attractive strategy to identify new immunotherapies from already approved drugs. Here, we developed a new platform to screen small molecule compounds based on a high-throughput luciferase-release cytotoxicity assay. We tested 1200 FDA approved drugs from the Prestwick Chemical Library, to identify compounds that increase NK cells’ cytotoxic potential. We found that the antibiotic colistin sulfate increased the cytotoxicity of human NK cells towards cancer cells. The effect of colistin was short lived and was not observed when NK cells were pretreated with the drug, showing how NK cell activity was potentiated only when the compound was present at the time of recognition of cancer cells. Further studies are needed to uncover the mechanism of action and the pre-clinical efficacy of colistin sulfate in mouse cancer models.

## 1. Introduction

Seminal studies from the 1990s and early 2000s highlighted the importance of the immune system in tumor biology through processes such as immunosurveillance and immunoediting [1], which in turn led to the most recent advances in cancer therapeutics: immunotherapies. Cancer immunotherapy is an encompassing term referring to therapeutic strategies that target components of the immune system to enhance clearance of the malignant cells. Various categories of immunotherapies exist [2] and, encouragingly, some became first-line treatments in some cancer types [3,4]. As the field of immunotherapy continues to develop, we have gained a better understanding of how immune cells contribute to immunotherapy efficacy, for instance, Natural Killer (NK) cells. NK cells are innate lymphoid cells that play a crucial role in tumor surveillance and clearance [5]. The importance of NK cells in immunosurveillance is appreciated from the observation that mice without functional NK cells show impaired tumor control [6,7] and that patients with defective or a decreased frequency of NK cells are at greater risk of developing malignancies, specifically virally induced cancers [8,9]. Recently, a thorough systematic review and meta-analysis encompassing 15 solid cancer types found that NK cell infiltration in solid tumors was associated with improved overall survival [10], whereas a lower low frequency of circulating or tumor-infiltrating NK cells or NK cells that display an impaired function are associated with a worse prognosis in several cancer types [11,12,13,14]. NK cells can either be administered for adoptive cell therapy or can be directly targeted to enhance their anti-tumor activity [5]. Within the second category, small molecule drugs have been surprisingly overlooked, despite evidence that small molecules can potentiate NK anti-cancer functions [15,16]. In addition to enhancing cytotoxicity, small molecules can also be used to promote proliferation and maturation in expansion protocols for NK cells that are used for adoptive cell therapies [15]. Small molecule immunotherapies are advantageous as they are orally bioavailable, usually cost less than biological immunotherapies, can target both extracellular and intracellular components, and have a greater ability to penetrate through physiological barriers [17]. However, from small molecule identification to development of a lead drug compound, the clinical drug pipeline can take years if not decades before the drug sees use in the clinic. For this reason, drug repurposing is an attractive alternative that identifies new indications for previously approved drugs. Repurposed drugs already have a safety and efficacy profile associated with them which makes this a favorable route. Several studies have conducted high-throughput drug screenings to identify drugs that modulate NK cell activity. These studies utilized commercially available libraries containing repurposed drugs [18,19,20] or natural compound libraries [21,22] and have identified small molecules that were not previously known to modulate NK cell activity. Here, we screened the Prestwick Chemical Library for compounds capable of enhancing the cytotoxicity of human NK cells towards leukemia target cells and identified the antibiotic colistin sulfate as an enhancer of NK cell cytotoxicity.

## 2. Materials and Methods

### 2.1. Cell Culture

All cell lines were cultured in a humidified incubator at 37 °C and 5% CO_2_ in media supplemented with 100 U/mL penicillin (Gibco, San Diego, CA, USA), 100 μg/mL streptomycin (Gibco), 10 g/mL gentamycin sulfate (Gibco), and 20 mM HEPES (Fisher, Mississauga, ON, Canada). NK92 cells were cultured in RMPI-1640 containing 10% fetal bovine serum (FBS) (Gibco). K562-nano luciferase (NL) and Ramos-NL cells were cultured in RMPI-1640 containing 5% FBS. A375-NL cells and 786O-NL cells were cultured in DMEM (Corning, VA, USA) containing 10% FBS.

### 2.2. Reagents and Drugs

Preparation of coelenterazine substrate: 500 μg of coelenterazine substrate (CTZ) (Gold Biotechnology, MO, USA) was reconstituted in 610 μL of 100% ethanol and 6.2 μL of 12 N hydrochloric acid. The reconstituted substrate was protected from light and stored at −80 °C until use. Prior to measuring luciferase activity, the reconstituted CTZ was mixed with 1× salt buffer (45 mM EDTA, 30 mM sodium pyrophosphate, 1.425 M NaCl) at a 1:200 dilution (5 μL CTZ per 1 mL salt buffer).

The Prestwick Chemical Library (https://www.prestwickchemical.com, accessed on 7 June 2022) was kindly provided by Dr. Diallo.

Preparation of candidate drugs: nicotinamide (Sigma-Aldrich, St. Louis, MO, USA), monensin sodium salt (Sigma-Aldrich), zafirlukast (Sigma-Aldrich), tizanidine hydrochloride (Sigma-Aldrich), closantel (Sigma-Aldrich), benazepril hydrochloride (Sigma-Aldrich), and diflorasone diacetate (Sigma-Aldrich) were prepared at a master stock concentration of 1 mM in 100% DMSO, whereas colistin sulfate salt (Sigma-Aldrich) was dissolved in water. A working stock concentration was prepared for all candidate drugs of 100 μM in PBS with a final DMSO concentration of 10%. All candidate drugs were stored at −20 °C until use.

Fluorochrome-conjugated antibodies were all from BD Biosciences (San Jose, CA, USA) CD3 (Clone UCHT-1), CD4 (Clone RPA-T4), CD8 (Clone RPA-T8), CD56 (Clone B159), CD16 (Clone 3G8), CD19 (Clone SJ25-C1), CD14 (Clone MφP9), CD107a (Clone H4A3), and IFN-γ (Clone 4S.B3).

### 2.3. Generation of Cell Lines

To generate K562, A375, 78O, and Ramos cells expressing nanoluciferase, lentiviral particles were produced by co-transfecting 293T cells with a lentiviral plasmid encoding nano luciferase plenti-NL (a gift from Dr. Wanker through Addgene; http://n2t.net/addgene:113450, accessed on 7 June 2022; RRID:Addgene #113450), packaging plasmids pCMV-dR8.2dvpr (a gift from Dr. Weinberg through Addgene; http://n2t.net/addgene:8455, accessed on 7 June 2022; RRID:Addgene #8455), and pCMV-VSV-G (a gift from Dr. Weinberg through Addgene; http://n2t.net/addgene:8454, accessed on 7 June 2022; RRID:Addgene_8454), following Lipofectamine 3000 transfection instructions for a 10 cm dish (Invitrogen, San Diego, CA, USA). Seventy-two hours following the transfection, supernatant containing lentiviral particles was collected and used to transduce K562, A375, and 786O-NL cells by spin-infection (500 g for 2 h at 37 °C) with 8 μg/mL polybrene (Sigma-Aldrich). Four days post-transduction, nano luciferase expression was confirmed by using the FMZ luciferase assay system (Promega, Madison, WI, USA). After nano luciferase expression was confirmed, single cells from the transduced cell populations were sorted into five 96-well plates using the MoFlow XDP Cell Sorter (Beckman Coulter, Brea, CA, USA). After several weeks of culture, wells with cell growth were tested for luciferase expression. Selected clones were mixed at an equal ratio to make a polyclonal population.

### 2.4. Luciferase Release-Based Cytotoxicity Assay

NK92 cells were co-cultured with target cells expressing NL in triplicate at various E:T (effector:target) ratios in RPMI 5% FBS in 96-well V bottom plates (Sarstedt, Montreal, QC, USA) for 5 h at 37 °C. After the incubation, 50 μL of supernatant from each well was transferred to round-bottom black 96-well plates (Corning, Kennebunk, ME, USA). Depending on the experiment, either 25 μL of FMZ substrate or CTZ substrate was added to each well and the Biotek Synergy Mx plate reader (Biotek, Winooski, VT, USA) was used to measure luminescence. Percentage (%) specific lysis was calculated using the following equation (Equation (1)), where experimental release is the raw luminescence values from NK92 + target cells, spontaneous release is the raw luminescence values from the target cells in absence of effector cells, and maximal release is the raw luminescent values from target cells treated with 30 μg/mL of digitonin (Sigma-Aldrich, St. Louis, MO, USA).
(1)$$\%\;\mathrm{specific}\;\mathrm{lysis}=\frac{\left(\mathrm{experimental}\;\mathrm{release}-\mathrm{spontaneous}\;\mathrm{release}\right)}{\left(\mathrm{maximal}\;\mathrm{release}-\mathrm{spontaneous}\;\mathrm{release}\right)}\times 100$$

### 2.5. Screening of the Prestwick Chemical Library and Plate Configuration

The Prestwick Chemical Library, which contains 1200 regulatory-approved drugs, was screened to identify compounds capable of enhancing NK92 cytotoxicity. K562-NL cells alone or a co-culture of NK92 + K562-NL cells at an E:T ratio of 1 were treated with 10 μM of each drug for 5-h at 37 °C. Each compound was evaluated in singlet over 2 independent experiments.

The Prestwick Chemical Library’s 15 stock plates were stored at −20 °C in 10% DMSO at a concentration of 100 μM in deep well plates (Axygen, Tamaulipas, Mexico), with compounds only in columns 2–11. On the day of the screen, the stock plates were thawed, and the Bravo Automated Liquid Handling Platform (Agilent, Santa Clara, CA, USA) was used to dispense 10 μL of each drug (final drug concentration of 10 μM) to columns 2–11 to a total of thirty 96-well V-bottom assay plates (Sarstedt, Newton, NC, USA). For the 15 assay plates containing K562-NL cells alone, 45 μL of K562-NL cells plus 45 μL of media was dispensed to all columns for a final assay volume of 100 μL. For the 15 assay plates containing K562-NL + NK92 cells, 45 μL of K562-NL was dispensed to all columns and 45 μL of NK92 cells was dispensed from column 1–11 for a final assay volume of 100 μL. Controls were dispensed in column 1 and 12 for each assay plate. For all 30 assay plates, 10 μL of 10% DMSO was dispensed to column 1 (final DMSO concentration of 1%). For the 15 assay plates containing K562-NL cells alone, 10 μL of 300 μg/mL of digitonin was dispensed to column 12 (final concentration of 30 μg/mL) as a maximal release control. For the 15 assay plates containing K562-NL + NK92 cells, NK92 and K562-NL cells were dispensed at a 9:1 E:T ratio in column 12 as a positive control for NK92 cytotoxicity. Negative controls in column 1 were K562-NL + DMSO (K562-NL alone plates) and 1:1 E:T + DMSO (NK92 + K562-NL plates). Plate layouts are depicted in Appendix A.

After the incubation, 50 μL of supernatant from each assay plate was transferred to round-bottom black 96-well plates (Corning, Kennebunk, ME, USA) using the Bravo Automated Liquid Handling Platform. Biotek plate reader was used to dispense 25 μL of CTZ to each well and measure luminescence.

### 2.6. Z′ Factor, Fold-Change, and Normalization Analysis

Raw luminescence values from the screenings were used to calculate Z′-factor and luminescent fold-change. To evaluate the overall screening assay stability, Z′ factor was calculated for each assay plate using the negative (column 1: DMSO) and positive (column 12, 9:1 E:T or digitonin) control values in each plate. Equation (2) was used to calculate *Z′* factor, where *SD*^+^ represents the standard deviation of the positive control, *SD*^−^ represents the standard deviation of the negative control, *m*^+^ represents the mean of the positive control, and *m*^−^ represents the mean of the negative control.
(2)$$Z^{\prime }=1-\frac{3SD^{+}+3SD^{-}}{\left|m^{+}-m^{-}\right|}$$

To identify compounds capable of enhancing NK92 cytotoxicity, the luminescent fold-change over DMSO control was calculated for all compounds in the E:T = 1 condition and K562-NL alone condition. Compounds that had a fold-change ≥1.3 were considered drug hits. Drugs were excluded if the fold-change was ≥1.3 in the K562-NL alone, treated with drugs, condition. Fold-change for each plate was calculated by using the controls on individual plates.

### 2.7. Dose–Response Experiments

K562-NL cells alone or NK92 and K562-NL cells mixed at a 1:1 and 3:1 E:T ratio were treated with either 0, 1, 5, 10, and 20 μM of drug candidates for 5 h at 37 °C. Luciferase activity was measured as previously described.

### 2.8. Colistin Sulfate Pre-Treatment

NK92 or K562-NL cells alone were treated for either 24 h or 1 h with 10 μM of colistin sulfate. Prior to mixing the cells together, the drug was washed out. Pre-treated NK92 cells were mixed with untreated K562-NL cells, and pre-treated K562-NL cells were mixed with untreated NK92 cells. A condition where drug treatment was present during the 5-h incubation was also included. Luciferase activity was measured as previously described.

### 2.9. Human PBMC Isolation

Human blood samples were obtained from healthy donors using the Perioperative Human Blood and Tissue Specimen Collection Program protocol approved by The Ottawa Health Science Network Research Ethics Board (OHSN-REB 2011884-01H). PBMCs were isolated from peripheral blood of healthy donors by Ficoll (GE Healthcare, Danderyd, Sweden) gradient centrifugation at 19 °C. Cells were resuspended in CryoStor^®^ CS10 (BioLife Solutions, Bothell, WA, USA) and cryopreservation of cells was followed according to the manufacturer’s instructions (BioLife Solutions, Bothell, WA, USA).

### 2.10. Flow Cytometry-Based Functional Assays

NK92 cells were stimulated with tumor cells for 5 h in the presence of 1 μg of Golgi Plug (BD), 1 μg of Golgi Stop (BD), 100 U of recombinant human IL-2 (Roche, Basel, Switzerland), and CD107a mAb antibody [23].

### 2.11. Human NK Cell Cytotoxicity Assays with Colistin Treatment

Previously frozen human PBMCs from healthy donors were thawed according to CryoStor^®^ CS10 (BioLife Solutions, WA, USA) thawing cells protocol (BioLife Solutions, WA, USA), and afterwards, PBMCs were kept overnight at 4 °C. NK cells were isolated from PBMCs using EasySep™ Human NK Cell Isolation Kit (Stemcell Technologies, Vancouver, BC, Canada). NK cells were co-cultured with K562-NL cells at different E:T ratios and treated with 10 μM of colistin for 5 h at 37 °C. Luciferase activity was measured as previously described.

### 2.12. Statistical Analysis

Statistical analyses conducted included two-tailed Mann–Whitney test and one or two-way ANOVA with either Dunnett’s, Sidak, Tukey’s, or Bonferroni’s multiple comparison test, as described in the figure captions. Statistical significance was achieved when the *p*-value was≤
0.05. GraphPad Prism 9 (GraphPad Software, Inc., San Diego, CA, USA) was used for statistical analyses. For flow cytometry experiments, FlowJo V.10.7.1 was used for analysis.

## 3. Results

### 3.1. Generation of K562-NL and Validation of a Luciferase Release-Based Killing Assay

Traditional methods to assess NK cell cytotoxicity, such as chromium-release or flow cytometry-based assays, are difficult to scale up for a high-throughput use. Luciferase release-based killing assays have proven useful to perform drug screenings [24], and a luciferase released-based screen was employed in a previous NK cell drug screening [18]. To generate target cells suitable for a luciferase release-based NK cell killing assay, we transduced the myeloid leukemia cell line K562 with a lentiviral plasmid encoding nano luciferase (NL). The expression of NL was assessed on transduced K562 cells by exposing cellular lysates to the substrate: no signal was observed from the lysate of control cells, whereas a robust signal was detected from the lysate of transduced K562 cells (Appendix A). Once we verified the NL expression on K562 cells, we employed K562-NL as targets in a luciferase release-based killing assays using the NK cell line NK92 as an effector. Consistent with what is expected in cytotoxicity assays, the presence of effector cells increased the luminescence signal in a dose-dependent manner, indicating that the target cells were effectively killed, whereas the luminescence signal observed with target cells alone was similar to that of the media only (Figure 1A).

Next, we sought to obtain a polyclonal population of K562 cells expressing NL from the transduced population, which likely contained cells which were not transduced. Therefore, we sorted single cells into five 96-well plates and tested wells where cell growth was observed for luciferase expression. K562 clones expressing NL were then tested in cytotoxicity assay vis-à-vis with the unsorted K562-NL population (Appendix A). Six clones that were killed by NK92 cells similarly to the K562 bulk population were selected and mixed at an equal ratio to make a polyclonal population of K562-NL cells, which was then used in all subsequent experiments.

### 3.2. Optimization of the Conditions for a High-Throughput Luciferase Release-Based Cytotoxicity Assay

For these first experiments, to test NL activity, we used furimazine (FMZ), the optimized substrate for NL [25] commonly found in commercially available kits (e.g., Nano-Glo, Promega, Madison, WI, USA). However, using FMZ in a high-throughput setting is not feasible due to the high cost of the substrate. Therefore, we explored if the less expensive substrate coelenterazine (CTZ), widely used for Renilla and Gaussia luciferase, could be used as an alternative. After conducting a luciferase release-based cytotoxicity assay, we used either FMZ or CTZ to assess NL activity side-by-side. A luminescent signal was detected with both substrates, although the magnitude of luminescence was higher using FMZ (Figure 1B). However, the dynamic range between targets only and the E:T ratio of 1 was comparable between the two substrates, and CTZ maintained the same dose-response observed using FMZ, indicating that CTZ could effectively replace FMZ as a substrate for these experiments.

Next, we set to determine the ideal E:T ratio to use for the drug screening. We conducted several luciferase release-based cytotoxicity assays that included a range of E:T ratios and chose to use an E:T ratio of 1 as this ratio shows minimal killing, but still has detectable luminescence above K562-NL target cells alone, and there is large dynamic range between the 1 and 81 ratios, an E:T ratio that shows saturation in killing (Figure 1C).

As a positive control for the screen, we decided to use the mild detergent digitonin, as we found it able to effectively lyse targets cells without compromising NL activity (Figure 1D).

For these set-up experiments, the supernatant from the luciferase-release cytotoxicity assay was collected and transferred to a new plate after a centrifugation step, which would be hardly feasible in high-throughput conditions. Our concern was that by skipping the centrifugation step prior to collecting the supernatant, we would capture live target cells that would lyse after the addition of the substrate, resulting in a similar luminescence detection between target cells alone and target + effector cells. Therefore, to determine if this step was required, we tested the difference between directly collecting the assay’s supernatant at the end of the cytotoxicity assay with or without a centrifugation step. To our advantage, the difference between the target cells alone and target + effector cells condition was still retained without the centrifugation step (Figure 1E). Based on these results, we deemed that a centrifugation step prior to supernatant collection was unnecessary and decided to proceed with directly collecting the assay supernatant for the drug screening.

Finally, to optimize the high-throughput drug screening workflow, we needed to estimate if leaving the effector or target cells at room temperature for an extended amount of time before they were seeded would affect the results as, logistically, we could not keep cells in their cell culture conditions (humidified incubator, 37 °C, 5% CO_2_) when seeding the drug screening assay plates. We simulated drug screen plating conditions by incubating NK92 and K562-NL cells separately at room temperature for 0, 60, 120, 180, and 240 min before the cells were seeded into assay plates. We observed that leaving the cells at room temperature for more than 120 min before being seeded into assay plates gradually, but substantially decreased NK92 cytotoxicity (Figure 1F). We also observed a slight increase in spontaneous lysis in the K562-NL alone condition as time progressed, shown by the increase in luminescence detection at the last two time points (Figure 1F). Based on these results, we concluded that cells had to be seeded within 1-h to maintain the dynamic range between the experimental and control conditions.

### 3.3. Screening of the Prestwick Chemical Library to Identify Enhancers of NK Cell Cytotoxicity

To identify compounds that enhanced NK cell cytotoxicity, we employed the Prestwick Chemical Library. K562-NL cells alone or NK92 + K562-NL cells mixed at an E:T ratio of 1 were treated with 10 μM of each drug for 5 h at 37 °C (Figure 2A). Each compound was evaluated in singlet over two biological replicates. To identify compounds that increased NK92 cytotoxicity, the luminescent values of all wells containing drugs were compared to the DMSO control wells from the same plate and this difference was quantified as a fold-change over the DMSO control (Figure 2B). Fold-change values for all compounds and the list of excluded compounds are listed in Appendix A.

Compounds with a fold-change ≥1.3 were considered to have increased NK cell cytotoxicity. We identified 87 drugs that had a fold-change ≥1.3 from the first screening of the Prestwick Chemical Library and 119 drugs that had a fold-change ≥1.3 from the second screening. From this list, only Alexidine dihydrochloride proved to be toxic for target cells. even in absence of effectors, and was, therefore, not further considered. Fourteen compounds from the total drugs identified had a fold-change ≥1.3 on both screening days (Table 1). From these fourteen drugs, eight candidate drugs were selected for follow-up experiments. Drugs with a higher fold-change were prioritized and drugs that were no longer in use, not available in the North American market, or were already known to be enhancers of NK cytotoxicity were excluded. The eight candidate drugs and associated fold-changes were colistin sulfate salt (2.02), nicotinamide (1.85), monensin sodium salt (1.62), zafirlukast (1.54), tizanidine hydrochloride (1.42), closantel (1.41), benazepril hydrochloride (1.40), and diflorasone diacetate (1.40) (Table 1).

To evaluate the overall screening assay stability, the Z′ factor was calculated for each assay plate from the screening of the Prestwick Chemical Library. The screening assay had an average Z′ factor of 0.72 for the K562-NL alone plates treated with drugs and 0.44 for the NK92 + K562-NL (E:T of 1) plates treated with drugs. A Z′ factor close to 0.5 is considered fair and a Z′ factor of 0.5–1 is considered good [26]. A Z′ factor for each individual plate can be found in Appendix A. Z′ factor analysis suggests that the overall quality of the drug screening was fair.

### 3.4. Validation of Candidate Drugs

The initial validation of the eight candidate drugs was conducted by performing cytotoxicity assays following the drug screening experimental conditions (E:T ratio of 1 and drug concentration of 10 µM). Of the eight drugs we tested, only colistin sulfate salt (herein colistin) increased NK cell cytotoxicity (Figure 3A), whereas the other seven drugs did not change, or even reduce, the ability of NK cells to kill target cells (Figure 3B–H).

Validation was expanded over two E:T ratios (1 and 3) and over a wider range of drug concentration (1–20 µM). Colistin was effective starting from 5 µM at both E:T ratios (Figure 4A), and increased cytotoxicity even at lower E:T ratios (Figure 5A). In contrast, the other compounds failed to elicit NK cell cytotoxicity in the tested conditions (Figure 4B–H), indicating they were likely false positives. Notably, colistin did not only increase the NK-mediated killing of K562 cells, but also of another cell line of a hematopoietic origin: the B cell lymphoma Ramos (Figure 5B).

Both K562 and Ramos cells are widely used to study NK cell cytotoxicity due to their high susceptibility to NK recognition and killing. Given the promising results obtained with colistin, we tested if this compound would also increase the NK-mediated killing of more resistant cell lines. For these studies, we employed the melanoma cell line A375 and the renal adenocarcinoma cell line 786O, both transduced with NL. Whereas the killing of K562 was potentiated by drug treatment, neither A375 nor 786O cells were killed more effectively in the presence of colistin sulfate (Figure 6 and Appendix A), indicating that the compound failed to generally boost NK cell cytotoxicity. On the other hand, colistin increased IFN-γ production in NK cells exposed to 786O cells, but not K562 cells, whereas no IFN-γ was detected in NK cells stimulated with Ramos or A375 cells (Figure 7B).

Taken together, these results show that colistin improves the activity of NK cells against several cell lines of a different origin.

### 3.5. The Effect of Colistin Sulfate on NK Cells Is Short Lived

To gather insights on the mechanisms underlying colistin-enhanced NK cell killing, we pre-treated NK92 cells with the drug for 24 h and then employed them as effectors in killing assays. NK92 cells were pre-treated with 0, 1, 5, or 10 μM of colistin and, prior to incubation with K562-NL cells, the drug was washed out. Consistent with the results of our screening, colistin was not toxic towards NK92 cells (Figure 8A). However, NK92 cells pre-treated with colistin failed to kill target cells more effectively than the control, whereas, consistent with what is described above, colistin increased the NK-mediated killing when present during the co-culture (Figure 8B).

Considering that NK92 cells pre-treated for 24-h with colistin did not present increased cytotoxicity, we tested if shorter pre-incubations could be more effective. NK92 cells were treated with 10 μM of colistin for 1 h, and the drug was washed out prior to co-culture with K562-NL cells. In comparison to the untreated condition, 1-h pre-treatment of NK92 cells with colistin slightly, but consistently increased the NK92 cytotoxicity (Figure 8C). On the other hand, pre-treatment of K562 cells did not increase the NK cell killing (Figure 8D), suggesting that colistin sulfate, rather than sensitizing the target cells, acted on NK cells, but in a short-lived fashion. On the other hand, colistin acted very quickly on NK cells, as the improved killing of the cancer cells was observed as early as 2 h in the culture (Appendix A).

### 3.6. Colistin Sulfate Increases Cytotoxicity of Primary Human NK Cells

Lastly, we tested if colistin treatment increased cytotoxicity of primary NK cells. We obtained PBMCs from healthy donor blood and isolated NK cells by negative selection. NK cell purity was ~85% and both CD56^+^CD16^−^ and CD56^+^CD16^+^ NK cell populations were present (not shown). After NK cells were isolated from PBMCs, they were immediately co-cultured with K562-NL cells at varying E:T ratios and treated with 10 μM of colistin for 5 h. Pooled results from all three healthy donor are shown in Figure 9. Consistent with the results obtained using NK92 cells, primary NK cells treated with colistin showed an increased cytotoxicity towards K562-NL.

Overall, this screen of the Prestwick Chemical Library identified colistin as a potential enhancer of NK cell cytotoxicity towards some cell types. The effect of colistin sulfate was short-lived, but was observed in both NK92 and primary human NK cells.

## 4. Discussion

Here, we conducted a high-throughput luciferase release-based cytotoxicity assay to screen the Prestwick Chemical Library and identify compounds that increased NK cell cytotoxicity. Luciferase-release assays have been used in a previous NK cell drug screening with success [18]. The results of that screening identified small molecule inhibitors of NK cell cytotoxicity; however, no small molecule enhancers were identified. To our knowledge, this is the first drug screening to employ a luciferase-release cytotoxicity assay format that identified small molecules capable of increasing NK cell cytotoxicity. Overall, the quality of our screening assay was fair, as determined by the Z′ factor. One problem we faced was that the dynamic range between the <1:1 E:T DMSO negative control> and <9:1 E:T positive control> in the 1:1 E:T plates decreased overtime, which is most likely due to decreased cytotoxicity of NK92 cells as time progressed. In preparation for the screen, we realized that cells kept at room temperature for more than 2-h prior to co-culture for the cytotoxicity assay showed decreased cytotoxicity, and therefore, separated the screen in two days. However, the signal of the <9:1 E:T positive control> began to overlap with the signal of the <1:1 E:T DMSO control> and, as a result, reduced the Z′ factor. To improve future screenings, a strategy to maintain a low signal-to-noise ratio overtime will be needed. Another limitation of the drug screening was that, due to a COVID-19-related shortage of plastic material, each compound was only tested in two biological replicates, which prevented us from performing a more robust statistical analysis of our results.

However, even in light of these limitations, the screening identified compounds previously reported to enhance NK cell activity, including amphotericin B [19]. We also identified nicotinamide, which is currently under investigation in clinical trials as a supportive agent for the ex vivo expansion of primary NK cells for the treatment of Non-Hodgkin lymphoma and multiple myeloma [27]. Interestingly, nicotinamide failed to be validated in follow-up experiments at the tested concentrations. On the other hand, we likely had several false negative hits, for example, two compounds identified in previous screens, naftifine and butenafine [20], were not highlighted in our screening (fold-change 0.84 and 1.14, respectively). In addition, we identified monensin, a known inhibitor of NK-cell degranulation as an enhancer of NK92 cytotoxicity. As expected, upon further tests, monensin was shown to decrease NK92 cytotoxicity in a dose-dependent manner. False positive and false negative hits, while problematic, are hard to avoid in medium-size screens such as ours. As for all screens, only further repetition and corroboration, potentially by other groups, will help better clarify whether candidate drugs have the desired effect.

From the eight candidate drugs identified from the screening of the Prestwick Chemical Library, colistin was the only drug that increased NK cell cytotoxicity in validation experiments. Colistin, also known as polymyxin E, is an antibiotic derived from *Bacillus polymyxa* and is used to treat antibiotic-resistant infections [28]. Colistin is an amphipathic molecule that disrupts the membrane of Gram-negative bacteria by displacing calcium and magnesium ions [28]. This ultimately leads to increased cell permeability and eventually cell death. Few studies have investigated how colistin modulates immune cells, let alone NK cells. One study found that colistin increased the cytotoxicity of murine splenic NK cells towards YAC-1 target cells and increased the production of IFN-γ [29]. Although no mechanism was provided, it was shown that both polycationic peptide and hydrophobic tail domains were needed for the observed effect [29]. Colistin was also shown to increase NK cytotoxicity in combination with the antibiotics daptomycin or teicoplanin in a mouse model of a multi-resistant *Acinetobacter baumannii* infection, and this combination showed a greater increase in NK cytotoxicity than either antibiotic on its own [30]. Colistin was previously identified in a Prestwick Chemical Library screen as a compound capable of enhancing the p38/MAPK pathway, a key pathway in innate immunity that is induced from TLR signaling [31,32]. Subsequently, this group showed that colistin increased phagocytosis, cytokine secretion, and phosphorylation of p38 in rat macrophages, and the KEGG pathway analysis of treated macrophages showed the upregulation of genes involved in signal transduction, immune pathways, and calcium signaling [33]. Interestingly, colistin induced the upregulation of genes downstream of the MAPK and PI3K-Akt pathways [33] in conditions similar to those of our screen. The MAPK and PI3K-Akt pathways are implicated in the downstream signaling of NK-activating receptors [34], suggesting a potential mechanism for colistin increasing NK cytotoxicity and IFN-γ production. Altogether, these highlighted studies illustrate that colistin has potential immunomodulatory properties beyond its bactericidal activity.

The fact that the effect of colistin was short-lived and evidence that some drugs disrupting membrane integrity have been shown to increase exocytosis of lytic granules of NK cells [20], suggest that colistin facilitated granule exocytosis in NK cells. However, if this was the mechanism of action, we would expect that colistin would increase NK-cell-mediated killing towards all tested target cell lines, which was not the case. Moreover, colistin promoted increased IFN-γ production in NK cells only in response to 786O cells, suggesting a context-dependent effect of the compound. Therefore, the mechanism of action of colistin remains to be elucidated.

## 5. Conclusions

In conclusion, we first optimized a luciferase release-based NK cell cytotoxicity assay for a high-throughput format. Using this assay format, we screened the Prestwick Chemical Library for small molecules that had the ability to increase NK cell cytotoxicity and identified colistin sulfate salt as an enhancer of NK cell cytotoxicity.

## Figures and Tables

**Figure 1 cancers-14-02832-f001:**
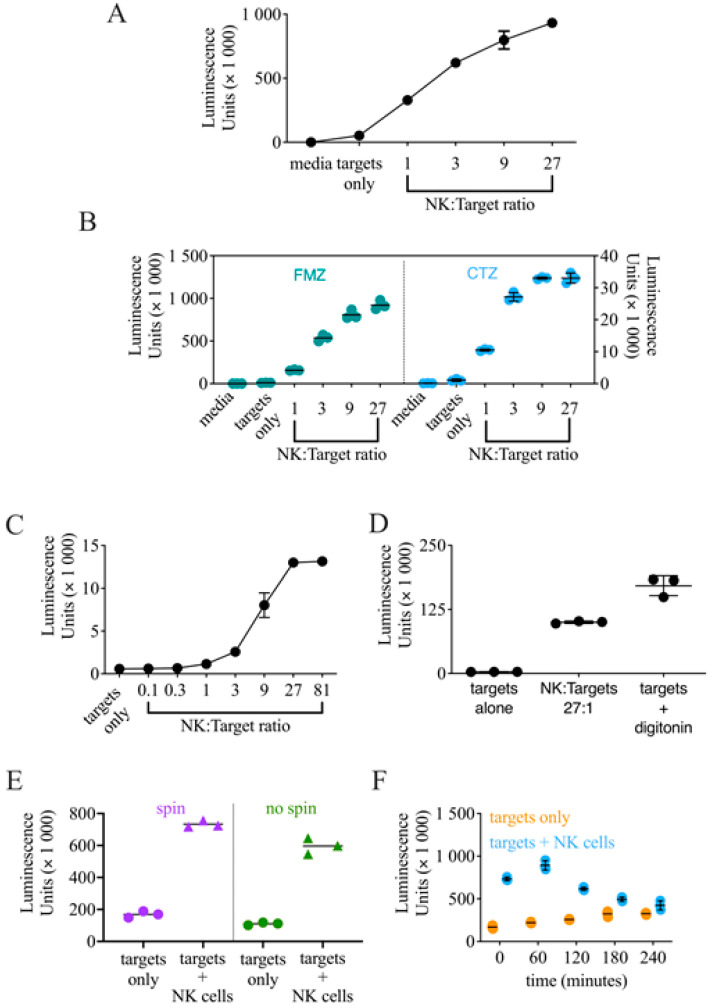
Optimization of a luciferase release-based cytotoxicity assay for a high-throughput screen. NK92 cells were co-cultured with K562-NL cells at the indicated E:T ratios for 5-h at 37 °C. After incubation, the supernatant was collected, the indicated substrate added, and luminescence read by Biotek Synergy (Winooski, WI, USA) microplate reader. (**A**) Luciferase-release cytotoxicity assay using transduced K562-NL cells (bulk population, unsorted). Twenty thousand K562-NL targets per well. After the incubation, luciferase activity was measured using FMZ (Promega Nano-glo, Madison, WI, USA) luciferase assay system. Mean +/− *SD* of three technical replicates. (**B**) After the incubation, luciferase activity was measured after addition of either FMZ luciferase assay system or CTZ substrate. Ten thousand K562-NL targets per well. Mean +/− *SD* of three technical replicates. Representative of 3 biological replicates. (**C**) Luciferase-release cytotoxicity assay using 10,000 K562-NL cells. After the incubation, luciferase activity was tested after addition of CTZ. Mean +/− *SD* of three technical replicates. Graph is representative of 2 biological replicates. (**D**) K562-NL cells were treated with 30 μg/mL digitonin and compared to K562-NL cells co-cultured at an E:T ratio of 27. Ten thousand K562-NL cells per well. Luciferase activity was tested after addition of CTZ. Mean +/− *SD* of three technical replicates. Representative of 2 biological replicates. (**E**) Supernatant from a cytotoxicity assay was either collected directly or collected after a centrifugation step. Three technical replicates are shown. Representative of 2 biological replicates. (**F**) NK92 or K562-NL cells were incubated at room temperature at indicated times (minutes) prior to start of a cytotoxicity assay. Mean +/− *SD* of three technical replicates. Representative of 3 biological replicates.

**Figure 2 cancers-14-02832-f002:**
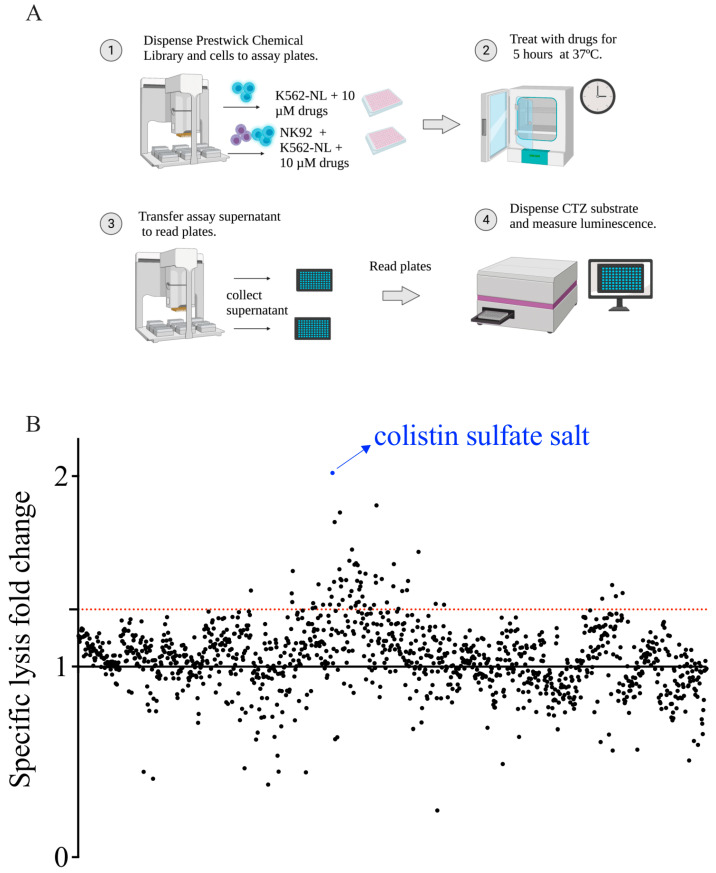
Screening of the Prestwick Chemical Library. (**A**) Schematic workflow of the Prestwick Chemical Library screening. (**B**) Average luminescent fold-change over DMSO control of all 1200 compounds from NK92 + K562-NL condition are plotted (each drug is represented by a black dot). The screening was conducted in singlet, over two biological replicates. The dotted red line indicates 1.3 fold-change. Compounds with a luminescent fold-change over DMSO control ≥1.3 were considered enhancers of NK92 cytotoxicity. Points represent the average fold-change of n = 2 biological replicates, except points 321–400 and 481–560 which represent fold-change from one replicate.

**Figure 3 cancers-14-02832-f003:**
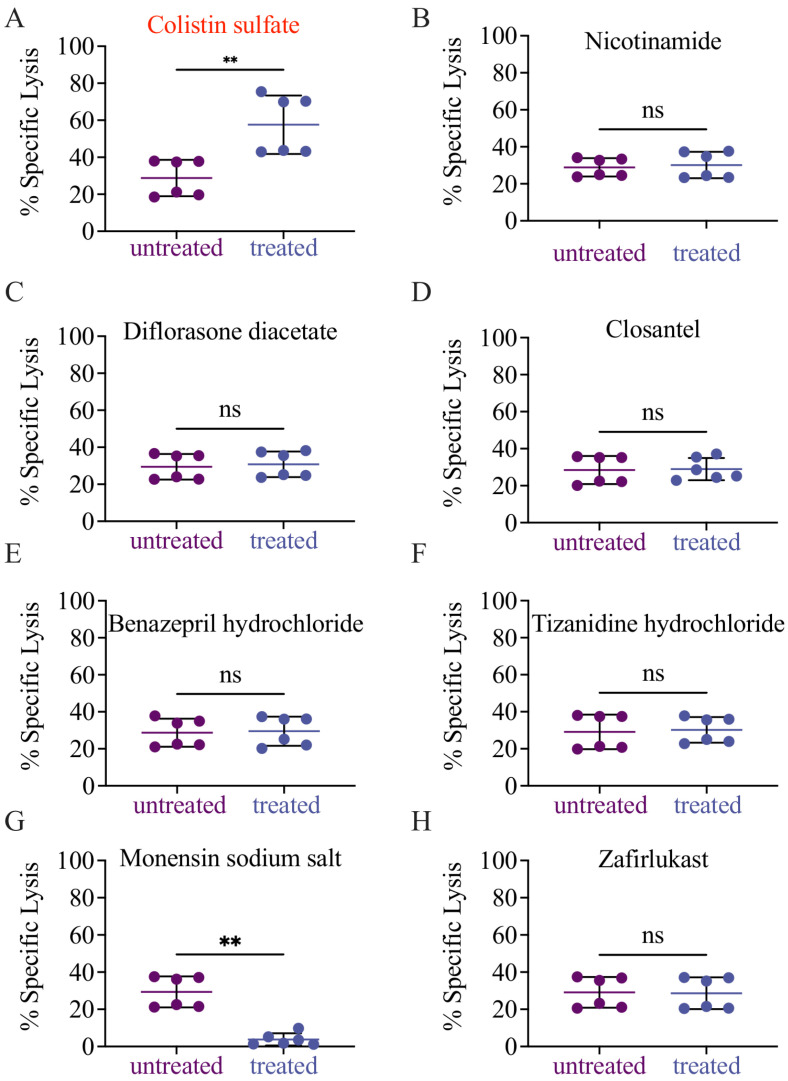
Colistin sulfate salt increases NK92 cytotoxicity against K562-NL. NK92 and K562-NL cells were seeded at an E:T ratio of 1 into 96-well V bottom plates containing either identified drug hits (10 μM) or DMSO for 5-h at 37 °C. Each drug tested had an independent untreated control. After incubation, the supernatant was subjected to a luciferase release assay. (**A**) Colistin sulfate salt. (**B**) Nicotinamide. (**C**) Diflorasone diacetate. (**D**) Closantel. (**E**) Benazepril hydrochloride. (**F**) Tizanidine hydrochloride. (**G**) Monensin sodium salt. (**H**) Zafirlukast. Each dot represents a technical replicate, experiments were repeated twice. Two-tailed Mann–Whitney test. ns = not significant, **: *p* < 0.01.

**Figure 4 cancers-14-02832-f004:**
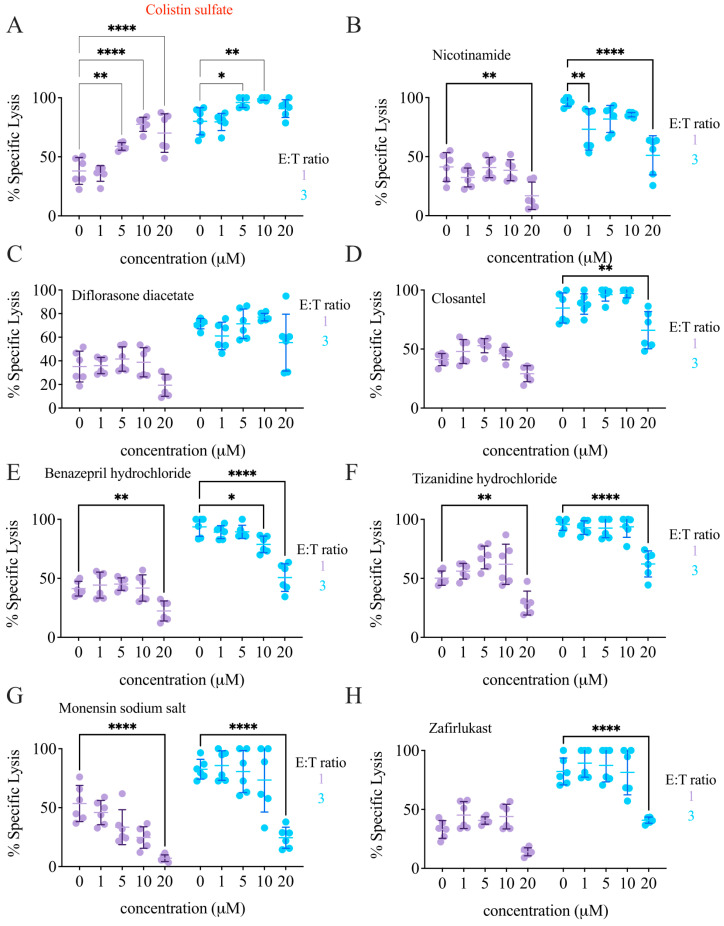
Colistin sulfate salt increases NK92 cytotoxicity against K562-NL. NK92 and K562-NL cells were seeded into a 96-well V bottom plate at a 1:1 or 3:1 E:T ratio with 10,000 K562-NL cells per well and treated with 1, 5, 10, and 20 μM of the indicated drug for 5-h at 37 °C. After incubation, the supernatant was subjected to a luciferase release assay. (**A**) Colistin sulfate salt. (**B**) Nicotinamide. (**C**) Diflorasone diacetate. (**D**) Closantel. (**E**) Benazepril hydrochloride. (**F**) Tizanidine hydrochloride. (**G**) Monensin sodium salt. (**H**) Zafirlukast. Each dot represents a technical replicate, experiments were repeated twice. Two-way ANOVA with Tukey’s multiple comparison test. *: *p* < 0.05, **: *p* < 0.01, ****: *p* < 0.0001.

**Figure 5 cancers-14-02832-f005:**
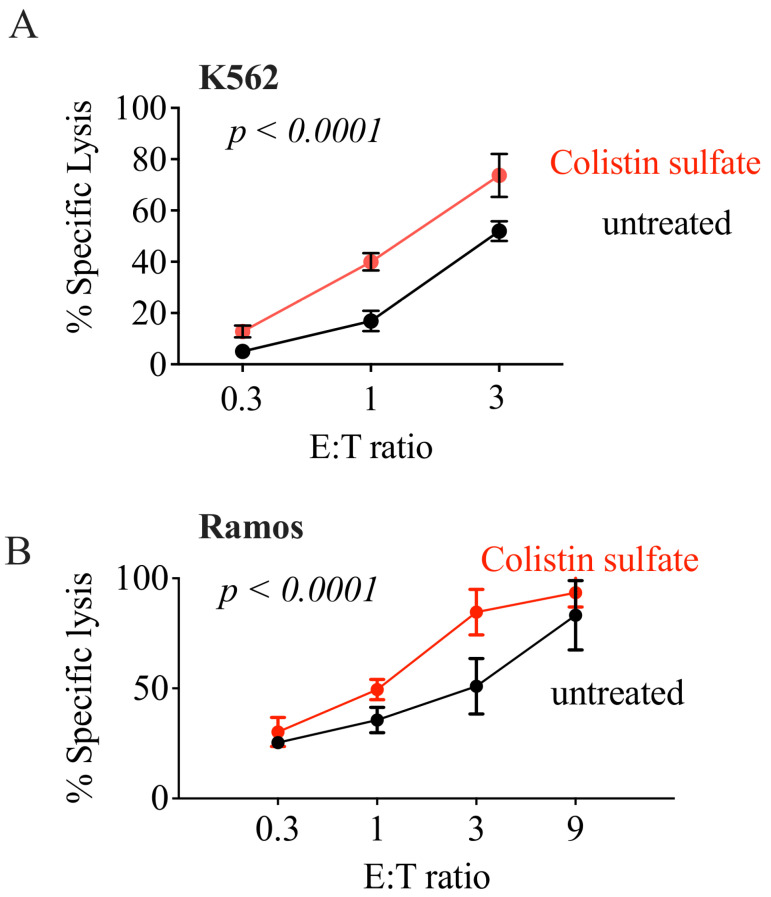
Treatment with colistin sulfate increases NK92 cytotoxicity against K562 and Ramos cells. NK92 cells were mixed with K562-NL (**A**) or Ramos-NL (**B**) cells and seeded into a 96-well V bottom plate at a 0.3:1, 1:1, and 3:1 E:T ratio with 10,000 K562-NL cells per well and treated with 10 μM of colisltin for 5-h at 37 °C. After incubation, the supernatant was collected and transferred to a 96-well black plate. The Biotek Synergy microplate reader (Winooski, WI, USA) was used to dispense CTZ and read luminescence. Percent specific lysis is depicted. Mean +/− *SD* of three technical replicates, the experiment was repeated twice. Statistical analysis with two-way ANOVA.

**Figure 6 cancers-14-02832-f006:**
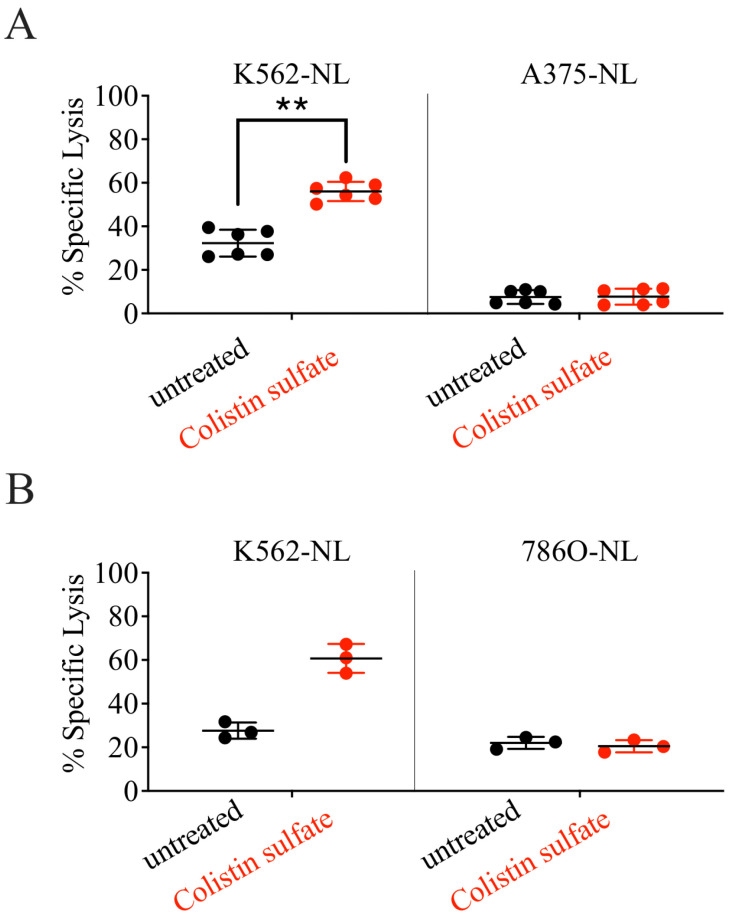
Treatment with Colistin sulfate failed to increase NK92-mediated killing of non-hematopoietic cell lines. NK92 and K562-NL/A375-NL (**A**) or K562-NL/786O-NL (**B**) cells were seeded into a 96-well V bottom plate at a 1:1 E:T ratio with 10,000 target cells per well and treated with 10 μM of colistin for 5-h at 37 °C. After incubation, the supernatant was collected and transferred to a 96-well black plate. The Biotek Synergy microplate reader (Winooski, WI, USA) was used to dispense CTZ and read luminescence. Percent specific lysis is depicted. Each dot represents a technical replicate, experiments were repeated twice. Two-tailed Mann–Whitney test. **: *p* < 0.01.

**Figure 7 cancers-14-02832-f007:**
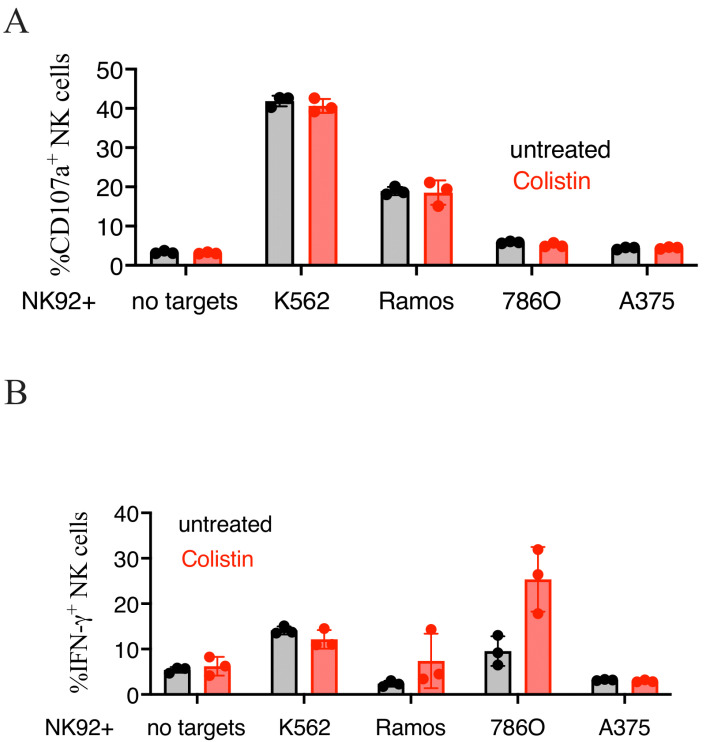
Treatment with Colistin sulfate increased IFN-γ production of NK92 stimulated with 786O cells. NK92 cells were stimulated at a 1:1 ratio with K562, Ramos, 786O, or A375 cells in the presence or absence of 10 μM of colistin for 5-h at 37 °C. After incubation, NK cells were stained for CD107a externalization (**A**) and IFN-γ production (**B**). Technical replicates of one experiment, representative of three performed, are shown.

**Figure 8 cancers-14-02832-f008:**
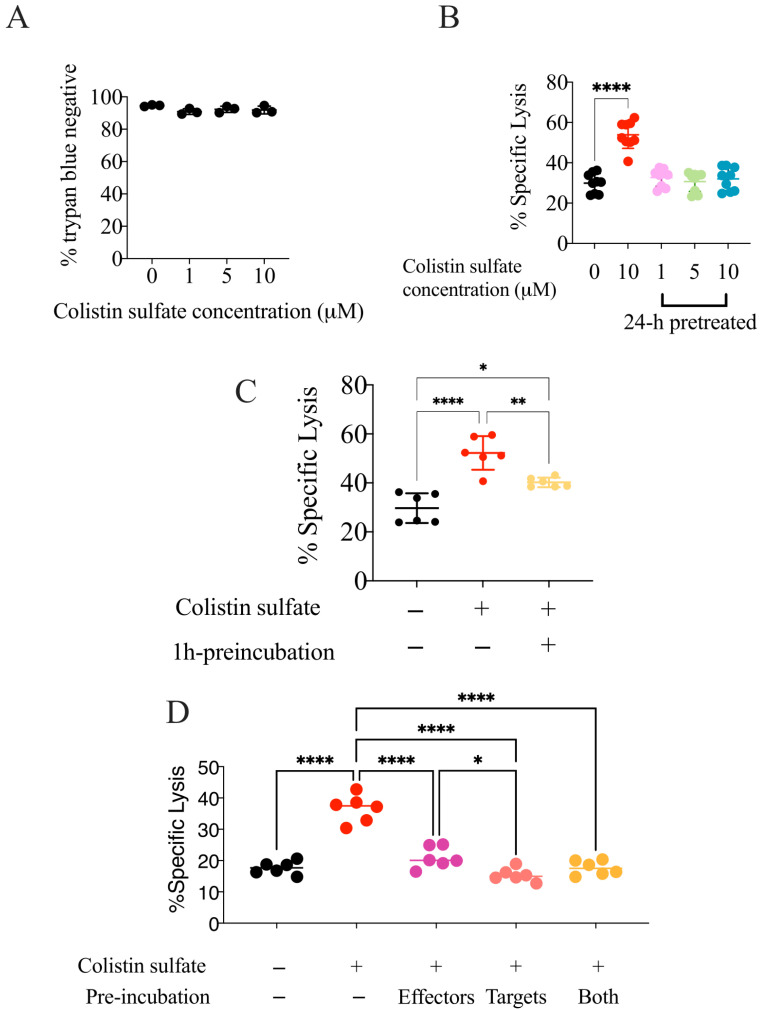
The effect of colistin sulfate on NK cells is short lived. (**A**) Percent viability of NK92 cells by trypan-exclusion dye after 24-h treatment with 0, 1, 5, and 10 μM of colistin. (**B**) NK92 cells were treated for 24-h with 0, 1, 5, and 10 μM of colistin. After 24-h, NK92 cells were washed and co-cultured with 10,000 K562-NL cells at a 1:1 E:T ratio for 5-h at 37 °C. NK92 cells co-cultured with targets cells and treated with 10 μM of colistin during the 5-h incubation were also included. After incubation, the supernatant was subjected to a luciferase release assay. The Biotek Synergy microplate reader (Winooski, WI, USA) was used to dispense CTZ and read luminescence. Percent specific lysis is depicted. Each dot represents a technical replicate, experiments were repeated three times. One-way ANOVA with Dunnett’s multiple comparison test. ****: *p* < 0.0001. (**C**) NK92 cells were treated for 1-h with 10 μM of colistin. After 1-h, the compound was washed and NK92 cells were co-cultured with 10,000 K562-NL cells in a 96-well V bottom plate at a 1:1 E:T ratio for 5-h at 37 °C. NK92 cells co-cultured with targets cells and treated with 10 μM of colistin during the 5-h incubation were also included. Percent specific lysis is depicted. Each dot represents a technical replicate, experiments were repeated twice. One-way ANOVA with Tukey’s multiple comparison test. ns = not significant, *: *p* < 0.05, **: *p* < 0.01; ****: *p* < 0.0001. (**D**) NK92 and K562-NL cells were pre-treated separately with 10 μM of colistin. After 1-h, cells were washed and co-cultured with either treated or untreated cells. Cells were seeded in a 96-well V bottom plate at an E:T of 1 ratio, using 10,000 K562-NL cells per well for 5-h at 37 °C. NK92 cells co-cultured with targets cells and treated with 10 μM of colistin during the 5-h incubation were also included. Percent specific lysis is depicted. Each dot represents a technical replicate, experiments were repeated twice. One-way ANOVA with Tukey’s multiple comparison test. Only significant differences between treatments are shown. *: *p* < 0.05; ****: *p* < 0.0001.

**Figure 9 cancers-14-02832-f009:**
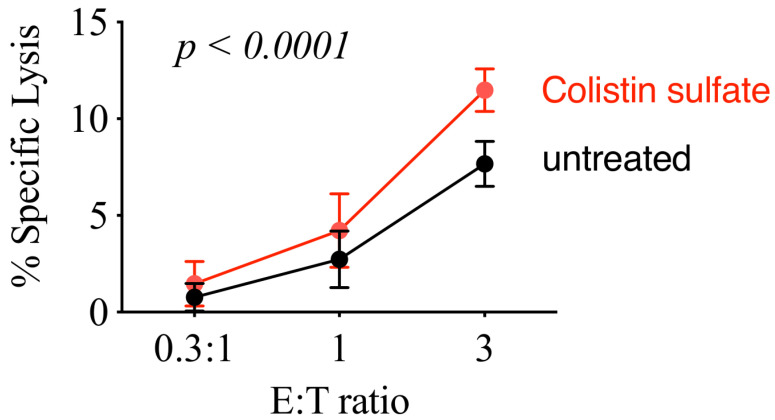
Colistin sulfate enhances the killing activity of primary human NK cells. Human NK cells isolated from PBMCs from healthy donors were mixed with K562-NL cells and were seeded into a 96-well V bottom plate at increasing E:T ratios, with 10,000 K562-NL cells per well, and treated with 10 μM of colistin for 5-h at 37 °C. After incubation, the supernatant was collected and transferred to 96-well black plates. The Biotek Synergy microplate reader (Winooski, WI, USA) was used to dispense CTZ and read luminescence. Percent specific lysis is shown. Mean +/− *SD* of three technical replicates, the experiment was repeated three times. Statistical analysis was conducted with two-way ANOVA.

**Table 1 cancers-14-02832-t001:** Compounds identified as enhancers of NK92 cytotoxicity from screening the Prestwick Chemical Library. Listed drugs had a luminescent fold-change ≥1.3 on both screening days. Drug class for each compound is listed. ^a^ Candidate drugs that were selected for further investigation. ^b^ Fold-change of single screening.

	Drug	Fold-Change	Drug Class
1	Colistin sulfate salt ^a.b^	2.02	antibiotic
2	Nicotinamide ^a^	1.85	vitamin B3
3	Monensin sodium salt ^a^	1.62	antibiotic
4	Butirosin disulfate salt	1.60	aminoglycoside antibiotic
5	Zafirlukast ^a^	1.54	anti-asthmatic
6	Amphotericin B	1.50	antifungal
7	Argatroban	1.46	anti-coagulant
8	Dimethisoquin hydrochloride	1.45	anesthetic
9	Tizanidine hydrochloride ^a^	1.42	adrenergic agonist
10	Closantel ^a^	1.41	anti-parasitic
11	Benazepril hydrochloride ^a^	1.40	ACE inhibitor
12	Diflorasone diacetate ^a^	1.40	topical steroid
13	Butoconazole nitrate	1.38	anti-fungal
14	Etretinate	1.34	retinoid

## Data Availability

The data presented in this study are available in Appendix A.

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
