# Peer review of "A New Functional Screening Platform Identifies Colistin Sulfate as an Enhancer of Natural Killer Cell Cytotoxicity"

_cancers, 2022, doi:10.3390/cancers14122832_

Round 1

Reviewer 1 Report

thank you for the the current version my feeling is that you increased the interest of your research I have to say that unfortunately I don't have enough time to read the document in a very precise way as during the first review but it appears to me that the article is now totally acceptable for publication...

Reviewer 2 Report

The authors have addressed all comments.

Reviewer 3 Report

The authors have adequately addressed my concerns. I particularly appreciate the additional experiments performed. In my view, the added data strengthen the work.

Reviewer 4 Report

The authors have satisfactorily answered all my comments.

This manuscript is a resubmission of an earlier submission. The following is a list of the peer review reports and author responses from that submission.

Round 1

Reviewer 1 Report

Thank you for this very interesting paper!!

As the title explains to the reader the aim of the paper seems to be the description of the activity of colistin sulphate on NK cells. However, the first experiments are devoted to the optimization of the screening. This part is of major importance and this information is not clearly introduce. May I suggest thinking about a change in the title of the article and in the summary with sentences explaining the experimental approach especially useful for people working on NK cells? Finally, the development of the experimental procedures seems more major than the interest for the results on colistin?

Line 49 “Small molecule immunotherapies are advantageous as they are orally bioavailable, usually cost less than biological immunotherapies, can target both extracellular and intracellular components and have a greater ability to penetrate through physiological barriers » I fully agree with you, unfortunately colistin is not orally available as far as I know and injection formulation is required… so I guess that this sentence can be maintained only if you precise somewhere that orally administration with colistin must be developed…

Line 294 you select 8 compounds out of 14, can you explain the reason why?

In table 1 I’m not sure to understand the sentence “b Fold-change of single screening » concerning only the colistin sulfate? can you explain this point to me?!

Figure 5 is not enough clear for me…  how many experiments and how many individual cultures in each experiment? it is of particular importance as the dose effect must be demonstrated carefully… (same question for figure 6 and 7… each point correspond to one experiment, one individual cell culture ???)

Line 426: to be clear I need some explanation!! « Pooled results from all 3 healthy donors are shown in Fig. 8. » three different experiments in triplicate for example? or one experiment including the 3 healthy donors?...

Conclusion: a lot of work in a very interesting domain. However, the final article is confusing for me… is it an article showing the optimisation of a screening or an article showing the major activity of an original compound as presented in the title?

My feeling is that for the first option this paper must be submitted in a Journal devoted to new protocols and for the second option the activity of colestin appears a little bit « short » for a publication in the current journal. I suggest modifying the summary and conclusion to be more balanced between the two options

Reviewer 2 Report

This using a novel drug screening approach to identify compounds that enhance NK92 cytotoxicity of K56 cells. The screening approach is very thorough and the results are potentially very interesting for the field.

Minor points:

Figures 3, 4 & 5 practically have the same title. I understand the authors are trying to be precise, but Is it really necessary to repeat all the dilutions with all the other drug candidate once we know they don’t enhance like colistin. It can be a bit tedious looking at data that’s not really that important to the central story.

Some figures and data points are a bit too small e.g. Fig 1 and Fig 2A. It would really enhance the paper if these could be enlarged. Use of colour can also sometimes be difficult to perceive e.g. Fig 1B.

Major points:

I may have missed this, but how can we be sure that colistin alone doesn’t enhance cell death of K562 in the absence of effector NK92 during the 5 hours of treatment at 37 degrees? It would be good to include this specific control to show that the colistin concentrations (1-20uM) used e.g. in Fig 4A don’t directly induce cell death of K562. Apologies if I’m wrong, but the authors appear to have included this control for NK92 cells using colistin concs 1-10uM in Fig 7. but not for K562? If not, this control really should be included.

Reviewer 3 Report

Cortés-Kaplan and colleagues report on the adaption of a luciferase release-assay of NK92-mediated cytotoxicity to screen a small compund  library in HTS mode for enhancers of this activity. This part of the study is very well done and described. They follow up with testing 8 identified candidate compounds and explore variation of experimental conditions for validated colistin sulfate, finding the effect to depend on target-effector cell interaction and to be restricted to only one of 3 cancer cell lines tested, the highly susceptible K-562 cells. They also confirm the effect to exist for human blood NK cells but do not further explore mechanisms (e.g., cytokine release, degranulation).

The study is overall thoroughly conducted and very well described, but I have major comments on some aspects of the Methods section and the validation part of the Results as well as on the Discussion.  Some minor comments follow below.

Major Comments

  1. On line 81 and the following, colistin is listed among the candidate drugs prepared in DMSO, and on lines 85/86 it is called an exception from this list and that it was prepared in water. In that case it should be removed from the DMSO list in the first place.
  2. In section 2.5, a flow cytometric assay is described. But all data presented in the mansucript appear to be be based on the luciferase release assay. Where does the reader find flow cytometry results? Also, Hsu et al. (2018) describe normalization of the target cell counts by deviding them by the bead counts. Here, the authors bpt the bead counts in the numerator instead. Then, for max. release, a value of zero target cells is assumed. In short, if this cannot be measured, and does not need to be measured, in the assyas strategy, there is no requirement for transforming the data into a "% specific lysis. For cell killing data by flow cytometry data, you just need to show the (normalized) target cell counts with and without effector cells. It would also be good to see exemplary dot plots of the FACS data for the gating strategy (removal of debrie and doublets, and gating in/out the live/dead target cells. But then again, where do I find FACS results in the first place?
  3. Section 3.2: I got confused with the introduction of furimazine. Please mention earlier in the Methods that this is the Nano-Glo substrate provided in the Promega kit and be consistent with how you refer to it.
  4. A general comment of the presentation of the data from Figure 3 onwards: Except for Figures 5 and 8, the use of different colors does not help, please stick to black and white. Unless you want to indicat that individual data points, e.g., for traeted and untreated in Figure 3, come from the same experiment. In Figure 3 for example, I see 6 data points for both conditions, and I assume these represent 6 biological repetitions (please be more clear about the numbers of technical and biological replicates each). So the data is then paired, which could be indicated by same colors (or in Figure 3 also by a connecting line).
  5. The issue of small numebrs of (apparently) paired samples leads to another point.  Given the small numbers of replcates in Figures 3-8, I believe no conclusion can be drawn on whether the data is normally distributed or not. Hence, a non-parametric test should be used for evaluating statistical significance of group differences, and for paired samples it should be a paired test. In the case of Figure 3 for instance the Wilcoxon signed-rank test. Possibly, get advise by an independent statistician.
  6. In the captions to Figures 3, 4, and for each panel in Figure 7, it is written that "the supernatant was collected and transferred to 96-well black plates". What for? I assume, for the luciferase release assay. You could simply say, supernatants were subjected to a luciferase release assay. An you need to mentio this only once in Figure 7 for all panels. 
  7. Figure 3: For each compound, the data pattern for the untreated condition looks very similar, but not identical. Question: Was an untreated condition done for every of the 8 candidate compounds separately, or do the patterns look slightly dissimilar for another reason? I would like to see clearly stated that the untreated condition was run 8 times, if so. In case it was one experiment for each replicate with a single untrated condition each, please make that clear.
  8. Figure 4: Several of the data points for the E:T ratio of 3 (blue) indicate a value of % specific lysis of over 100%. This is impossible! Please correct.
  9. In Figure 1A, the authors varied the E:T ration to make sure that in their screen they would have an appropriate dynamic range. But in Figure 6, they use the same E:T ratio of one as for K562 cells also for the two other cell lines. They do not show, whether they can see any killing at all of these cell lines by NK92 cells, even at higher ratios. So we do not know whether we are at all close to the lower limit of the dynamic range and could thus expect to see any improvement of killing by colistin as for K562 cells. Appropriate ratios should have been evaluated for A375 and 7860 cells also. 
  10. In Figure 7A, trypan blue dye exclusion data are shown as % viability. This assay is only detecting cells with already collapsed plasma membrane potential seen in cells at a relatively late stage of cell death. NK cells that are in early apoptosis, as for instance judged by annexin-V binding, are still trypan blue negativ, but I would not classify them as "viable" anymore. Hence, the y-axis label in Figure 7A should say"% trypan blue negative".
  11. In line 429/430 it is written that colistin was identified as a potential enhancer of NK cell cytotoxicity towards some cell types: I do not agree with "some cell cell types", because the effect  was only demonstrated for NK-killing of the highly susceptible K562 cells. As mentioned in comment 9, the results for two other cancer cell lines were not conclusive. And what leads me to my comment on the Discussion, the effect of colistin by blood NK cells was comparatively weak, i.e., < 5% specific lysis at the 1:1 E:T ratio compared to up to 60% by NK92 killing and aleady up to 20 by untratd NK92 cells. This should be discussed. Do we expect such high valuues for NK92 over blood NK cells from the literature?
  12. Discussion: The restrictionof the colistin effect to K562 cells and the lower magnitude of the effect by blood NK cells versus NK92 cells lowers the probability that colistin will show a biological effect in an animal model, which should be discussed. Nevertheless, knowing the mechanism is of high interest. The authors made some steps into this direction with demonstrating dependency of the colistin effect on effector-target cell contact. But absence of an effect of NK preincubation with colistin argues against a transcriptional/translational effect as discussed for macropahges on lines 484 ff.
  13.  I imagine 2 ways to ad biological relevance to the results of this study: First, the authors could determine wether colistin improved NK cytokine production/seceretion duing the 5-hour coincubation with target cells (possibly all 3 cell lines). Second, the authors could do a flow cytometry CD107a degraulation assay to assess whether A375 and 7860 cell resist NK92 killing because of impaired degranulation or bcause of target cell intrinsic deficiency to activate cell death pathways induced by NK cell derived granzyme B and FAS receptor activation.

Minor comments

  1. Section 2.1: Clarif the nomenclature for your modified cell lines, i.e., that  the NL following the cell line abbreviation stnads for nano luciferase.
  2. Line 109: How many clones were pooled? The term "polyclonal" suggests that these were much more than the 9 clones shown in supplementary Figure 2B. I ist rather"pauciclonal"?
  3. Today, the Prestwick Chemical Library is said on their web site to consist of 1,520 compounds. This study used 1,200. was that a subset or has the library grown since?
  4. Su. Fig. 2 is lacking a caption.
  5. Line 282/283 is a repetition from line 138/139.
  6. Line 456 ff: I would not expect high robustness of statistical analysis in the primary screen. But maybe it would be good to indicate the fold change in Table 1 for each of the two screen individually to get a feeling for the varince with the candidates. Colistin was found in only one screen. So I understand, that luck was involved which is OK. But I would not be comfortable with saying that COVID-19 related shortage of plastic consumabls prevented you from doing another screen. This work seems to be very recent then, and what has prevented you to wait a few more weeks for the next shipment of plates, vials and tips, or flasks hatever it was. At worst, you create the impression that the study was rushed.  

Reviewer 4 Report

In this study, Cortés-Kaplan et al describe the screening of the Prestwick chemical library, containing 1200 FDA-approved compounds, to identify molecules that increase NK cell cytotoxic potential. The screen is based on the measurement of NK cell mediated killing of a target cell line (K562) expressing the nanoluciferase. Instead of primary NK cells, the authors chose to perform their screen on the NK92 cell line. Out of the 1200 drugs tested, they obtained 14 hits that increased NK92 cytotoxicity by a factor >1.3. They further tested 8 of these 14 molecules and validated only one: Colistin sulfate. Six had no effect and one (Monensin) even had a negative effect. In addition, Colistin sulfate did not improve NK92 cytotoxicity toward two other cell lines (A375 and 786O). They further show that Colistin sulfate need to be present at the time of NK92-target cell encounter to enhance the killing. Of note, Colistin sulfate also increases primary human NK cell cytotoxicity toward K562 targets.  

The screen is well designed and very neatly described. It is however puzzling that out of the 8 hits further tested only one is validated. Below are some points that could addressed to improve the quality of the manuscript.

Out of the 8 candidates selecting for further testing, only one is validated. Is it expected following this type of medium-scale screen? In the same direction, the authors report that two previously identified molecules (naftifine and butenafine) were not identified in their screen suggesting that the results presented contains also false negative. Could the authors comment on this fact?

On Figure 2B, the authors present the raw data obtained. Some sort of pattern is observed (in particular for plates 1, 2, 3, 14 and 15. How the compounds were they ordered? According to their position in the plate? If so, this suggests that an “edge-effect” is at play in some plates. Could the authors comment on this?

What is the impact of Colistin sulfate on NK cell viability at the end of the co-culture with their targets?

What is the impact of Colistin on IFN-y secretion by NK cells upon stimulation?

Could the authors document the impact of Colistin on the kinetic of target cell killing upon exposure to NK92 or primary NK cells?